# Dietary Fiber Level Improve Growth Performance, Nutrient Digestibility, Immune and Intestinal Morphology of Broilers from Day 22 to 42

**DOI:** 10.3390/ani13071227

**Published:** 2023-03-31

**Authors:** Cheng Zhang, Erying Hao, Xiangyu Chen, Chenxuan Huang, Gengyun Liu, Hui Chen, Dehe Wang, Lei Shi, Fengling Xuan, Dongmei Chang, Yifan Chen

**Affiliations:** 1College of Animal Science and Technology, Hebei Agricultural University, Baoding 071001, China; 2Xiangda Hezhong Biotechnology Co., Ltd., Shijiazhuang 050800, China

**Keywords:** broiler, fiber level, growth performance, immune, intestinal morphology, nutrient digestibility

## Abstract

**Simple Summary:**

Dietary fiber is considered to be a nutrient concentration diluent and an antinutritional factor affecting poultry performance. As a consequence, commercial diets generally contain 2–3% crude fiber. However, crude fiber is also a functional component of normal digestive organ function. Moderate dietary fiber could promote the development of digestive organs, increase digestive enzyme activity and nutrient digestibility, improve health status and enhance growth performance in poultry. At present, there are few systematic studies on the dietary fiber requirements of broilers in the late feeding stage. Therefore, it is of great practical significance to determine the dietary fiber requirements of broilers. Under the conditions of this experiment, a diet with 7–9% crude fiber may promote growth performance by improving the nutrient digestibility, immunity and intestinal morphology of broilers from day 22 to 42.

**Abstract:**

There are few systematic studies on the dietary fiber requirements of broilers in the late feeding stage, and there are not enough data to support this hypothesis. This experiment was conducted to examine the effects of dietary fiber level on growth performance, nutrient digestibility, immune function and intestinal morphology of broilers from day 22 to 42. A total of 480 one-day-old Arbor Acres broilers with half male and half female were randomly allocated into four groups, with eight replicates in each group and fifteen chickens in each replicate. The experimental period was 42 days. All broilers were fed a basal diet from 1 to 21 days. During the 22–42 day period, the four experimental groups were fed diets with soybean hulls as the fiber source, and crude fiber (CF) levels were 2%, 5%, 8% and 11%, respectively. The results showed that during the 29–42 day period, the average daily feed intake (ADFI) of broilers was higher in the 5% CF and 8% CF groups (*p* < 0.05), and during the 29–35 day period, the average daily gain (ADG) of broilers was higher and the ratio of feed and gain (F/G) of broilers was lower in the 5% CF and 8% CF groups (*p* < 0.05). The digestibility of crude protein (CP), ether extract (EE), CF, acid detergent fiber (ADF) and neutral detergent fiber (NDF) was higher in broilers of the 8% CF group (*p* < 0.05). The immunoglobulin A (IgA), immunoglobulin G (IgG) and immunoglobulin M (IgM) content of the plasma of broilers was higher in the 8% CF group (*p* < 0.05). The villus height (VH) of the duodenum, jejunum and ileum of broilers was higher, and the crypt depth (CD) was lower in the 8% CF group than that in the 2% CF group (*p* < 0.05). The ratio of VH and CD (V/C) of the duodenum and jejunum of broilers in the 8% CF group was higher than that in the 2% CF group (*p* < 0.05). The quadratic regression analysis showed that the optimum dietary CF level was 7–9%. In conclusion, under the conditions of this experiment, a diet of 7–9% CF may promote growth performance by improving the nutrient digestibility, immunity and intestinal morphology of broilers from day 22 to 42.

## 1. Introduction

Dietary fiber has been considered a nutrient concentration diluent and an antinutritional factor affecting poultry performance [1,2,3]. The main reason was that insoluble dietary fiber had been used as a nutrient diluent due to the lack of enzymes to digest β 1-4, β 1-3 and β 1-6 linkages of non-starch polysaccharides [4], which would impair growth performance when the content was high [5]. As a consequence, commercial diets generally contain 2–3% crude fiber (CF) [6]. However, CF is also a functional component of normal digestive organ function [7,8,9]. Moderate dietary fiber could promote the development of digestive organs, increase digestive enzyme activity and nutrient digestibility, improve health status and enhance growth performance in poultry [2,10,11,12].

Dietary fiber has been associated with changes in growth performance [3,13,14,15,16,17], nutrient digestibility [7,18], intestinal morphology [11,18,19,20] and gastrointestinal regulation [2,21], which are generally ignored in animal diets, especially in poultry diets. The addition of 2% bagasse to the diet could increase the body weight of broilers [15]. Taylor et al. [22] reported that the average daily feed intake (ADFI) and feed conversion ratio (FCR) were linearly increased in the broilers fed the 15%, 30% and 45% oat hulls diets compared to control group. The addition of 3% oat hulls or sunflower hulls to low-fiber diets could improve average daily gain (ADG) and decrease the ratio of feed and gain (F/G) of broilers [14]. Supplementation with 1% or 2% lignocellulose [16] or 3% sunflower hulls [17] had no effect on the growth performance of broilers. Adding insoluble fiber, such as oat hulls, wood shavings and soybean hulls, could increase feed efficiency by 3–5% and increase body weight between 2–5% when included at 3–5% in broiler diets [23,24,25,26]. Subsequent studies reported the effects of dietary fiber on intestinal morphology. Feeding diets containing 8% CF with soybean hulls as a fiber source could increase the villus height (VH) of the duodenum in broilers aged from 1 to 20 days [18]. Supplementation with 0.5% or 1% inulin in the diet could increase the VH of the ileum in broilers [20].

Moreover, adding a 15% oat hull and barley hull mixture to the diet could increase the weight and volume of the gizzards of broilers [27]. Adding 1% insoluble fiber to the diet of broilers could increase the relative weight of the proventriculus, gizzard and liver and improve the proteolytic activity of the pancreas [28]. Broilers fed 3% wheat bran had the increased relative weights of their gizzards and small intestines and enhanced pancreatic amylase and trypsin activity, which was correlated with nutrient digestibility [29]. In addition, it has been reported that dietary fiber could regulate blood lipids and blood glucose and improve the health status of poultry [2,30]. Diets containing 4.7% CF with 30 g/kg rice hulls as a fiber source could increase antibody titers in broilers [31].

Against the background of precision feeding, it is of great practical significance to determine the dietary fiber requirements of broilers. However, at present, there are few systematic studies on the dietary fiber requirements of broilers in the late feeding stage. Therefore, the objective of this experiment was to determine the dietary fiber requirements of broilers in the late feeding stage by recording the effects of different dietary CF levels on the growth performance, nutrient digestibility, immune function and intestinal morphology of broilers.

## 2. Materials and Methods

All experiments were performed in accordance with the ARRIVE guidelines and were carried out in accordance with the UK Animals (Scientific Procedures) Act, 1986, and associated guidelines, EU Directive 2010/63/EU for animal experiments, and the National Institutes of Health’s *Guide for the Care and Use of Laboratory Animals* (NIH Publications No. 8023, revised 1978), approved by the Ethics Committee of Animal Experimentation of Hebei Agricultural University (Protocol 2023001).

### 2.1. Animals, Diets and Management

The experimental design was a single-factor test design. A total of 480 one-day-old Arbor Acres broilers, half male and half female, were in good health with similar body weights. These 1-day-old broilers were purchased from Hebei Jiuxing Farming Development Co., Ltd. The experiment lasted for 42 days. All broilers were fed a basal diet with 2% CF level from 1 to 21 days of age. At 22–42 days of age, all broilers were randomly allocated into 4 groups, with 8 replicates in each group and 15 broilers in each replicate. The four experimental diets were formulated to include different CF levels (2%, 5%, 8% and 11%) with soybean hulls as the fiber source. The analyzed compositions of the diets are summarized in Table 1. The diets were formulated in accordance with the NRC to meet the nutrient requirements of broilers [32]. Vitamins and minerals were provided in each diet to meet or exceed the nutrient requirements of broilers. During the first 7 days of the experiment, the temperature was maintained at about 33 °C and then gradually reduced to 23 °C. The chickens were exposed to light for 24 h during the first 7 days, and then the time of light exposure was gradually reduced to 16 h. Birds were kept in metabolic cages in one common room, and collection trays were installed in every cage [33]. Feed and water were provided ad libitum. The feeding, drinking, excreta and health condition of broilers were observed daily.

### 2.2. Growth Performance

During the 22–42 day period, broilers were weighed, and feed intake was recorded weekly. The ADG, ADFI and F/G were calculated.

### 2.3. Fecal, Blood and Tissue Collection

In order to determine in vivo nutrient digestibility, on day 40, two birds with body weights close to the mean of each group were randomly selected for a digestive trial. The acid-insoluble ash (AIA) marker technique was used to measure nutrient digestibility throughout the days 40–42. On day 40 (3 days before excreta collection), an additional source of AIA with celite was included in the diet at a dosage of 10 g/kg. On the last 3 days of the experiment, fresh fecal samples of the uppermost layer were collected 3 times in one day. Each 100 g fecal sample was added to 20 mL 10% hydrochloric acid to fix nitrogen. The fecal samples were dried to constant weight at 65 °C, crushed and stored in a sealed bag at −4 °C. At the end of the experiment (at 42 days), two broilers were randomly selected from each replicate, and 3 mL blood samples were collected from each broiler. After 30 min of standing in the anticoagulant tube, the supernatant was centrifuged at 3000 r/min for 10 min, and the supernatant was stored at −20 °C for further biochemical and immunological analyses. At the end of the experiment, two broilers were randomly slaughtered from each replicate. The bursa of Fabricius and spleen were removed, and the surface fat and connective tissue of the organs were removed. The blood stains were removed with filter paper. The proventriculus, gizzard, duodenum, jejunum, ileum and cecum were collected. The middle segments of 2–3 cm in the duodenum, jejunum and ileum were collected and fixed in 4% neutral buffered paraformaldehyde.

### 2.4. Nutrient Digestibility

Acid insoluble ash (AIA) was used as a digestibility marker to determine the apparent total tract digestibility (ATTD). The AIA, crude protein (CP), ether extract (EE), CF, acid detergent fiber (ADF) and neutral detergent fiber (NDF) in feed and excreta were measured by the AOAC method [34]. The digestibility was calculated using the following formula: ATTD (%) = (100 − A1/A2 × F2/F1 × 100), in which A1 represents the AIA content of the feed, A2 represents the AIA content of the feces, F1 represents the nutrient content of the feed, and F2 represents the nutrient content of the feces.

### 2.5. Immune Index

The contents of immunoglobulin A (IgA cat. NO. YX-090701C), immunoglobulin G (IgG cat. NO. YX-090707C) and immunoglobulin M (IgM cat. NO. YX-090713C) in plasma were determined by enzyme-linked immunosorbent assay (ELISA). All kits were purchased from Nanjing Jiancheng Bioengineering Institute (Nanjing, China). The bursa of Fabricius and spleen were accurately weighed and recorded. The immune organ indexes (bursa of Fabricius index and spleen index) were calculated as follows: immune organ index = organ weight (g)/body weight (kg).

### 2.6. Plasma Biochemical Index

An enzyme labeling instrument (Bio Tek Instruments, Inc., Vermont, VT, USA) was used to determine total cholesterol (T-CHO cat. NO. F002-1-1), triglyceride (TG cat. NO. A110-1-1), high-density lipoprotein (HDL cat. NO. A112-1-1), low-density lipoprotein (LDL cat. NO. A113-1-1), total blood glucose (GLU cat. NO. YX-071221C), growth hormone (GH cat. NO. YX-070800C) and insulin (INS cat. NO. YX-091419C) in plasma. All kits were purchased at Nanjing Jiancheng Bioengineering Institute (Nanjing, Jiangsu, China) and determined strictly according to the instructions.

### 2.7. Intestinal Morphology

After the washing, dehydration and clarification of the fixed intestine tissues, the samples were embedded in paraffin. Serial sections with a thickness of 5 mm were placed on a glass slide for dewaxing, hydration and staining. The VH and crypt depth (CD) were measured by a NIKON DS-U3 image processing and analyzing system (NIKON ECLIPE CI, Tokyo, Japan). The ratio of villus height to crypt depth (V/C) was calculated.

### 2.8. Gastrointestinal Development

The gizzard and proventriculus of broilers were weighed, and the organ indexes were calculated. The length and weight of the duodenum, jejunum, ileum and cecum were measured, and the relative length and relative weight of each intestine were calculated according to the following equations: organ index = organ weight (g)/body weight (kg); relative length = intestinal length (cm)/body weight (kg); and relative weight = intestinal weight (g)/body weight (kg).

### 2.9. Data Analysis

All data were analyzed by one-way ANOVA using SPSS software (version 23, SPSS Inc., Chicago, IL, USA), and the normality and homogeneity of variance were checked before statistical analysis. Each replicate was defined as an experimental unit for the trial. Statistical differences among treatment groups were compared using Duncan’s multiple comparison test. Polynomial contrasts were used to test the linear and quadratic response to the increasing levels of CF in diets. Quadratic regressions (Y = aX^2^ + bX + c) were fitted to the responses of the dependent variables to dietary CF supplementation levels. The extremum response for CF was defined as CF = −b/(2 × a). The results are presented as the means and standard error of the mean (SEM), and differences were considered significant at *p* < 0.05.

## 3. Results

### 3.1. Growth Performance

The effects of dietary CF level on the growth performance of broilers are shown in Table 2. The quadratic regression analysis between growth performance and dietary CF level is presented in Table 3. The ADFI of broilers in the 5% CF group and 8% CF group was significantly higher than that of broilers in the 2% CF group at 29–42 days of age (*p* < 0.05), and there was a significant quadratic correlation between the ADFI and CF level (*p* < 0.05). The ADG of broilers in the 8% CF group was significantly higher than that of broilers in the 2% CF group and 11% CF group at 29–35 days of age (*p* < 0.05), and there was a significant quadratic correlation between ADG and CF level (*p* < 0.05). The F/G of broilers in the 2% CF group and 11% CF group was significantly higher than that of broilers in the 5% CF group and 8% CF group at 29–35 days of age (*p* < 0.05), and there was a significant quadratic correlation between F/G and CF level (*p* < 0.05).

The indicators with significant quadratic correlation were selected for the optimal CF level analysis by the quadratic regression analysis of the correlation with dietary CF level (Table 3). The optimum CF levels for F/G and ADG were 7.00% and 7.08% at 29–35 days of age. The optimum CF levels for ADFI were 7.40% and 7.96% at 29–35 days of age and 36–42 days of age, respectively.

### 3.2. Nutrient Digestibility

The effects of dietary CF level on nutrient digestibility of broilers are shown in Table 4. The quadratic regression analysis between nutrient digestibility and dietary fiber level is presented in Table 3. The digestibility of CP, EE, CF, ADF and NDF in the 8% CF group was significantly higher than that in the other groups (*p* < 0.05), and the digestibility of CP, EE, CF, ADF and NDF in each group was significantly quadratically correlated with the CF level (*p* < 0.05). The optimum CF levels for CP, EE, CF, ADF and NDF digestibility in broilers were 8.61%, 8.66%, 9.13%, 7.50% and 6.29%, respectively.

### 3.3. Immune Index

The effects of dietary CF level on the immune function of broilers are shown in Table 5. The quadratic regression analysis between plasma immune function and dietary fiber level is presented in Table 3. The plasma IgA content of broilers in the 8% CF group was significantly higher than that in the other groups (*p* < 0.05). The plasma IgG content in the 8% CF group was significantly higher than that in the 2% CF group (*p* < 0.05). The plasma IgM content in the 8% CF group was significantly higher than that in the 2% CF group and 5% CF group (*p* < 0.05). There was a significant quadratic correlation between IgA, IgG, IgM and CF levels (*p* < 0.05). There was no significant difference in the immune organ index between groups (*p* > 0.05). The optimal CF levels for plasma IgA, IgG and IgM contents in broilers were 8.86%, 8.92% and 9.04%, respectively.

### 3.4. Plasma Biochemical Index

The effects of dietary CF level on plasma biochemical indexes of broilers are shown in Table 6. The quadratic regression analysis between plasma biochemical indexes and dietary fiber level is presented in Table 3. The T-CHO content of broilers in the 8% CF group and 11% CF group was significantly lower than that in the 2% CF group (*p* < 0.05), and there was a significant linear correlation between plasma T-CHO and CF levels in each group (*p* < 0.05). The GLU content of the 5%, 8% and 11% CF groups was significantly lower than that of the 2% CF group (*p* < 0.05), and the plasma GLU content of broilers in each group was significantly quadratically correlated with the CF level (*p* < 0.05). The GH of broilers in the 8% CF group was significantly higher than that in the 2% and 5% CF groups (*p* < 0.05), and there was a significant quadratic correlation between plasma GH content and CF level in each group (*p* < 0.05). There were no significant differences in other plasma biochemical indexes among the groups (*p* > 0.05). For the plasma GLU content and GH content of broilers, the optimal CF level was 8.70% and 9.03%, respectively.

### 3.5. Intestinal Morphology

The effects of dietary fiber level on the intestinal morphology of broilers are shown in Table 7. The VH of the duodenum, jejunum and ileum in the 8% CF group was significantly higher than that in the 2% CF group (*p* < 0.05). The CD of the duodenum, jejunum and ileum in the 8% CF group was significantly lower than that in the 2% CF group (*p* < 0.05). The VH and CD of the duodenum, jejunum and ileum in each group were significantly quadratically correlated with the CF level (*p* < 0.05). The V/C of the duodenum and jejunum of broilers in the 8% CF group was significantly higher than that in the 2% CF group (*p* < 0.05), and the V/C of the duodenum and jejunum in each group was quadratically correlated with the CF level (*p* < 0.05).

### 3.6. Gastrointestinal Tract Development

The effects of dietary CF level on the gastrointestinal development of broilers are shown in Table 8. The relative lengths of the ileum and cecum in the 8% CF group were significantly higher than those in the other groups (*p* < 0.05). There were no significant differences in the other gastrointestinal tract development indexes (*p* > 0.05).

## 4. Discussion

Improvements in intestinal morphology and organ development could increase nutrient absorption, which would enhance growth performance [19,27,28]. Adding an appropriate amount of CF to broiler diets could increase nutrient digestibility and immune function and improve intestinal health and the growth performance of broilers. The results of the current experiment showed that nutrient digestibility could be significantly improved when the CF level was 8% in broiler diets, and the ADFI of broilers in the late feeding period was significantly increased in the 5% and 8% CF groups. Oikeh et al. reported that the ADG/BW (body weight) were decreased in broilers fed a basal grower diet diluted with 0%, 5%, 10% or 15% lignocellulose [35]. Another study reported that the ADFI was increased with the diets dilution in broilers infected with Eimeria maxima, and the diets were a basal grower diet diluted with 0%, 5%, 10% or 15% lignocellulose [36]. The inclusion of fiber, such as oat hulls, wood shavings and soybean hulls, has been shown to increase feed efficiency by 3–5% and increase body weight by 2–5% when included at 3–5% in the broiler diets [23,24,25]. Another study reported that compared with the control group with 0 g/kg CF, the body weight gain and F/G of broilers in the diets with 24 g/kg CF and oat hulls as the fiber source were significantly increased and the F/G was significantly reduced [37]. Another study found that the body weight gain in the 10% dietary cellulose group was significantly lower than that in the 0% and 3.5% dietary cellulose groups [7]. Thus, when the fiber content of the diet exceeded the optimum value, nutrient absorption is interrupted and growth performance is negatively affected.

In the current study, the ADG of broilers first increased and then decreased at 29–35 days of age as the dietary CF level of diets increased from 2% to 11%, and the F/G was the lowest in the 5% and 8% CF groups. According to the quadratic regression analysis, the optimal CF levels for ADFI, ADG and F/G were 7.40%, 7.08% and 7.00% at 29–35 days of age and 7.96% for ADFI at 36–42 days of age. In summary, dietary 7–8% CF level can improve growth performance of broilers, and the amount of CF added to the broiler diets in the late feeding stage is 7–8%, considering the growth performance.

The nutrient digestibility of diets is closely related to digestive tract development and body health in animals. Improving nutrient digestibility is helpful for improving production efficiency. Digestibility is an important indicator reflecting the degree of nutrient digestion and the growth performance of animals. One study found that compared to the basal diet with 29.9 g/kg CF without oat hulls, the diet with 34.9 g/kg CF and oat hulls as a fiber source could increase the digestibility of dry matter, organic matter, EE and CP in broilers [38]. Jimenez-Moreno et al. reported that compared with the basal diet group with 2.5% oat hulls and 8.4% total dietary fiber, the experimental group with 5% oat hulls and 10.1% total dietary fiber significantly improved nutrient digestibility in broilers [39]. This is consistent with the results of the current study. The addition of 8% CF in diets could promote nutrient digestibility in broilers, and excessive or low fiber would affect digestibility. The optimum CF levels for CP, EE, CF, ADF and NDF digestibility in broilers were 8.61%, 8.66%, 9.13%, 7.50% and 6.29%, respectively. Dietary fiber could reduce the number of goblet cells in the intestinal villi epithelium, which would reduce the goblet mucin content and the cavity barrier formed by goblet mucin, resulting in accelerating the passage of nutrients through the intestinal wall [40]. Dietary fiber could increase pancreatic enzymatic activity and reverse peristalsis, which would lead to an increase in nutrient digestibility [2,23,25]. An appropriate amount of dietary fiber could improve the digestibility of nutrients. However, excessive dietary fiber would form a coating structure that reduce the accessibility of digestive enzymes to nutrients, thus interrupting the normal digestion [41,42]. Lower dietary fiber would reduce intestinal peristalsis and digestive enzyme, thus reducing nutrient digestibility [24]. In summary, in the current study, considering analysis results was that the optimum CF level of diets for nutrient digestibility was 8–9% in broilers aged 22–42 days.

In the current study, the digestibility of CP and EE in the 11%CF group was significantly higher than that in the 2% CF group but did not significantly affect F/G. The main reason may be that the nutrients first promote gastrointestinal development, which is that the VH and relative length of the ileum in the 11% CF group were higher than that in the 2% CF group in the current study. The gastrointestinal tract of broilers has a high demand for energy during digestion, with over 20% of the dietary available energy being first utilized in the intestinal mucosa. This form of energy is mainly derived from the oxidation of glucose and amino acids [43]. Therefore, high nutrient digestibility may not necessarily improve growth performance.

Immunoglobulin, as an important substance in humoral immunity, is produced by immune organs and can improve antibacterial, antiviral and cell phagocytosis functions [44]. Blood immunoglobulin is the main antibody of humoral immunity, including IgA, IgG and IgM, which are able to reflect the immune function of the body to a certain extent [45]. Immunoglobulins, especially IgA and IgG, enhance the immunity of chickens, and these antibodies are reactive to self or foreign antigens [45]. IgA can prevent and neutralize the colonization of exogenous invasive pathogens on the mucosal surface and is the main antibody of mucosal local immunity. The IgA concentration reflected local immunity. IgM is the immunoglobulin with the strongest early immune effect and is an important barrier against pathogen invasion. At the same time, IgM was also able to accelerate the production of IgG. IgG was the immunoglobulin with the highest concentration and participated in humoral immunity [46]. In the current experiment, the IgA, IgG and IgM contents of plasma of broilers in the 8% CF group were significantly higher than those in the 2% CF group, and the optimum CF levels of diets for IgA, IgG and IgM contents of plasma in broilers were 8.86%, 8.92% and 9.04%, respectively. In summary, adding 8–9% CF to the diets may have the best effects in terms of increasing the immunoglobulin content, and the immune function could be improved in broilers aged 22–42 days.

Plasma biochemical indexes are important indexes for evaluating metabolism and health status. The stability of GLU in plasma plays an important role in maintaining the activities of normal life. The changes in GLU in plasma reflected the dynamic balance of GLU absorption, transport and metabolism [47]. In the current experiment, the plasma GLU content of broilers in the 8% CF group was significantly lower than that in the other groups, and the optimal CF level of broiler diets for GLU content of plasma was 8.70%, but the difference in INS content in each group was not significant. The results were consistent with the study that dietary fiber could improve INS sensitivity of peripheral tissue, reduce the insulin requirements of the body and then decrease the blood GLU level but had no significant effect on insulin level [48]. The concentrations of T-CHO, LDL and HDL are important indexes that reflect the level of lipid metabolism. A study showed that 11% CF levels could improve blood lipid metabolism in broilers [49]. Similarly, 8% and 11% CF levels in broiler diets significantly reduced the plasma T-CHO content in the current study. GH is an important hormone in the growth axis that is closely related to the metabolism of proteins, sugars and lipids and can improve growth and development in animals [50]. In the current experiment, the plasma GH content of broilers in the 8% CF group was significantly higher than that in the 2% and 5% CF groups, and the optimal CF level of broiler diets for plasma GH content was 9.03%. This showed that adding 9.03% CF to broiler diets could improve growth hormone secretion and promote the growth and development of broilers in the late feeding period. In summary, 8–9% CF of diets could reduce GLU concentration and maintain INS levels, improve lipid metabolism and increase GH secretion in broilers at the late feeding stage, thereby promoting the growth and development of broilers.

Poultry require a certain amount of dietary fiber for normal intestinal physiological functions [24]. The intestinal villous is the main site of nutrient absorption, and its development determines the absorption and utilization efficiency of nutrients [11]. VH could increase the surface area of epithelial cells in the intestinal tract in contact with chyme and promote intestinal absorption. However, when the villus became shorter, there were fewer mature cells and the nutrient absorption capacity was decreased [51]. The main function of intestinal crypts is to secrete digestive juices. The CD reflected the generation rate of cells. The crypt became shallow, which indicated that the maturation rate of cells and the secretion function increased [52]. Studies have shown that an 8% CF level diet with soybean hulls as the fiber source could significantly increase the VH of the duodenum, jejunum and ileum of broilers [18,53]. The VH of the duodenum and jejunum of laying hens fed 5% CF diets with 7.5% soybean hulls were improved compared with the control group fed the basal diet with 2% CF [54]. A further study showed that feeding isonitrogetic and isoenergetic diets with CF levels from 2.8% to 9% to turkeys resulted in an increase in the number and size of villi in all sections of the small intestine in groups fed a diet containing higher fiber levels [11]. Consistent with the above results, in the current study, the VH and V/C of the duodenum and jejunum and the VH of ileum of broilers in the 8% CF group were significantly higher than those in the 2% CF group. This might be associated with the reverse peristalsis caused by the stimulation of CF, which would promote villus development [27]. The CD of the duodenum, jejunum and ileum of broilers in the 8% CF group was significantly lower, and the V/C of the duodenum and jejunum of broilers in the 8% CF group was significantly higher compared with the 2% and 11% CF groups in the current study. The mechanism may be that short-chain fatty acids (SCFAs) produced by dietary fiber fermentation are able to stimulate the proliferation of intestinal cells and SCFAs are easily absorbed and utilized by intestinal cells [55,56]. The 8% CF diet improved intestinal morphology and promoted nutrient digestion.

The gastrointestinal tract (GIT) is the main site of nutrient digestion and absorption in the body. The mechanisms of fiber functions in the gastrointestinal tract depend on the chemical structure, particle size and concentration [13,57,58]. The development of the gastrointestinal tract could directly reflect the digestion and absorption function of the body. Adding fiber ingredients could dilute the diet and may improve gastrointestinal peristalsis. Adding 3% oat hulls and soybean hulls to the broiler diet could increase the proventriculus and gizzard size and improve feed conversion [24]. Chickens fed 3% wheat bran have increased relative gizzard weights and increased small intestine activity, which is correlated with nutrient digestibility [29]. A study reported that the relative weight of the proventriculus and the gizzard of ducks fed alfalfa meal as a fiber source increased significantly with increasing CF levels (16.7 g/kg, 42.6 g/kg, 77.9 g/kg and 101.6 g/kg, respectively) in the diet [59]. Insoluble dietary fiber, such as cellulose, lignin and arabinoxylans, are able to modulate the size of the small intestine, cecum and pancreas [10]. The results of the current study were similar to the above results. We found that the relative length of the ileum and cecum of broilers in the 8% CF group was significantly higher than that in the other groups, and the relative length of cecum in broilers was the largest at the CF level of 8.00%, according to the quadratic regression analysis. It can be observed that dietary fiber had the function of promoting gastrointestinal development within a certain range. The 8% CF diet was able to stimulate the development of the gastrointestinal tract in broilers aged 22–42 days and was beneficial to the digestion and absorption of nutrients, which promoted the healthy growth of broilers. This is also consistent with the results of the current study where the optimal amount of CF in diets for nutrient digestibility was 8–9% and 8% for intestinal morphology in broilers aged 22–42 days.

## 5. Conclusions

In summary, under the conditions of this study, 7–9% CF levels in the diets of broilers aged 22–42 days were able to improve immunity, gastrointestinal development and intestinal morphology and improve nutrient digestibility and growth performance in the late feeding period.

## Figures and Tables

**Table 1 animals-13-01227-t001:** Ingredient composition and nutrient levels of experimental diets from day 22 to 43 (dry matter basis).

Item	Crude Fiber Level, %			
	2%	5%	8%	11%
Ingredients, %				
Corn	70.76	61.54	51.74	42.41
Soybean meal	13.20	12.00	11.90	10.90
Soybean hull	0.30	7.90	15.40	22.90
Soybean oil	1.40	3.80	6.30	8.70
Soybean isolate protein	4.50	5.00	5.00	5.00
Corn protein powder	5.50	5.50	5.50	6.00
Limestone	1.26	1.15	1.03	0.92
Calcium hydrogen phosphate	1.62	1.67	1.71	1.76
Premix ^1^	0.50	0.50	0.50	0.50
L-Lys HCl (98.5%)	0.32	0.31	0.30	0.30
Sodium chloride	0.35	0.35	0.35	0.35
DL-Methionine	0.09	0.12	0.14	0.16
L-Threonine	0.10	0.06	0.03	0.00
Choline chloride (50%)	0.10	0.10	0.10	0.10
Total	100.00	100.00	100.00	100.00
Nutrient levels ^2^				
ME/(kcal/kg)	3103.00	3100.00	3098.00	3102.00
Crude protein	18.95	18.99	19.02	19.01
Crude fiber	1.99	5.00	8.01	10.98
Calcium	0.90	0.90	0.90	0.90
Available phosphorus	0.42	0.42	0.42	0.42
Acid detergent fiber	2.22	5.87	9.53	13.16
Neutral detergent fiber	7.57	11.91	16.27	20.61
Methionine	0.45	0.47	0.47	0.48
Lysine	1.10	1.11	1.12	1.12

^1^ The premix provided the following per kg of the diet: VA 10,000 IU, VD 30,000 IU, VE 15 IU, VK 3.0 mg, VB1 1.2 mg, VB2 7 mg, VB12 0.08 mg, VB6 6 mg, niacin acid 40 mg, biotin 0.15 mg, folic acid 1.0 mg, Cu 10 mg, Fe 60 mg, Zn 80 mg, Mn 80 mg, I 1 mg, Se 0.3 mg. ^2^ Except for protein and crude fiber, nutrients are calculated values.

**Table 2 animals-13-01227-t002:** Effects of dietary CF level on growth performance of broilers from day 22 to 42.

Day-Old	Item	Crude Fiber Level				SEM	*p*-Value		
		2%	5%	8%	11%		A	L	Q
22–28	ADFI, (g/bird/d)	83.23	85.82	81.56	83.17	0.72	0.219	0.826	0.976
	ADG, (g/bird/d)	39.29	43.33	41.67	38.43	1.84	0.088	0.323	0.586
	F/G (g/g)	2.17	2.02	2.13	2.20	0.11	0.069	0.347	0.425
29–35	ADFI, (g/bird/d)	107.25 ^b^	112.18 ^a^	112.60 ^a^	110.56 ^ab^	0.75	0.036	0.114	0.016
	ADG, (g/bird/d)	49.11 ^b^	53.57 ^ab^	55.36 ^a^	49.40 ^b^	1.15	0.024	0.249	0.001
	F/G (g/g)	2.30 ^a^	2.01 ^b^	2.04 ^b^	2.25 ^a^	0.06	0.012	0.202	0.001
36–42	ADFI, (g/bird/d)	118.65 ^b^	125.57 ^a^	127.42 ^a^	125.31 ^a^	1.16	0.028	0.244	0.012
	ADG, (g/bird/d)	70.48	74.29	72.50	74.29	1.72	0.293	0.258	0.531
	F/G (g/g)	1.60	1.80	1.75	1.75	0.47	0.348	0.305	0.426

In the same row, values with no letter or the same letter superscripts mean no significant difference (*p* > 0.05), while values with different small letter superscripts indicate significant differences (*p* < 0.05). A, *p*-value of one-way ANOVA; L, *p*-value of linear analysis; Q, *p*-value of quadratic analysis. The same as below. ADFI, average daily feed intake; ADG, average daily gain; F/G, ratio of feed and gain. The number of samples is 480.

**Table 3 animals-13-01227-t003:** Estimation of the optimal response for dietary CF levels based on quadratic regressions in broilers.

Dependent Variables	Regression Equation	R^2^	*p*	Optimal Response to CF Levels
ADFI (29–35 d), g/d	Y = −0.193 X^2^ + 2.856 X + 102.411	0.337	0.016	7.40%
ADG (29–35 d), g/d	Y = −0.389 X^2^ + 5.507 X + 35.971	0.476	0.001	7.08%
F/G (29–35 d), g/d	Y = 0.020 X^2^ − 0.280 X + 2.968	0.495	0.001	7.00%
ADFI (36–42 d), g/d	Y = −0.250 X^2^ + 3.978 X + 111.753	0.356	0.012	7.96%
GLU, mmol/L	Y = 0.031 X^2^ − 0.539 X + 6.802	0.676	0.001	8.70%
GH, μg/mL	Y = 0.036 X^2^ −0.650 X + 4.443	0.380	0.011	9.03%
IgA, μg/mL	Y = −1.179 X^2^ + 20.899 X + 116.144	0.645	0.001	8.86%
IgG, μg/mL	Y = −8.833 X^2^ + 157.622 X + 904.456	0.308	0.021	8.92%
IgM, μg/mL	Y = −2.934 X^2^ + 53.037 X + 446.116	0.460	0.002	9.04%
Relative length of cecum, cm/kg	Y = 0.018 X^2^ − 0.288 X + 446.116	0.400	0.001	8.00%
CP, %	Y = −0.369 X^2^ + 6.351 X + 20.468	0.594	0.003	8.61%
EE, %	Y = −0.598 X^2^ + 10.352 X + 14.727	0.897	0.009	8.66%
CF, %	Y = −0.334 X^2^ + 6.099 X + 2.914	0.759	0.001	9.13%
NDF, %	Y = −0.152 X^2^ + 1.911 X + 8.283	0.762	0.001	6.29%
ADF, %	Y = −0.110 X^2^ + 1.650 X − 1.231	0.517	0.006	7.50%

The optimal value was the maximum or minimum response to dietary CF levels according to each regression equation (%); R^2^, determination coefficient; P, *p*-value of quadratic effect; Y, dependent variable; X, dietary CF level (%). CP, crude protein digestibility; EE, ether extract digestibility; CF, crude fiber digestibility; NDF, neutral detergent fiber digestibility; ADF, acid detergent fiber digestibility.

**Table 4 animals-13-01227-t004:** Effects of dietary CF level on nutrient digestibility (%) of broilers on day 42.

Item	Crude Fiber Level				SEM	*p*-Value		
	2%	5%	8%	11%		A	L	Q
CP	33.81 ^c^	36.64 ^c^	53.99 ^a^	43.54 ^b^	2.07	0.015	0.069	0.003
EE	34.40 ^d^	47.24 ^c^	63.71 ^a^	55.22 ^b^	2.78	0.010	0.081	0.009
CF	15.23 ^d^	20.65 ^c^	34.70 ^a^	28.08 ^b^	1.96	0.012	0.130	0.001
NDF	11.78 ^c^	13.16 ^b^	14.70 ^a^	10.61 ^c^	0.41	0.019	0.608	0.001
ADF	2.16 ^d^	2.67 ^c^	6.56 ^a^	3.12 ^b^	0.45	0.017	0.090	0.006

CP, crude protein; EE, ether extract; CF, crude fiber; NDF, neutral detergent fiber; ADF, acid detergent fiber. In the same row, values with no letter or the same letter superscripts mean no significant difference (*p* > 0.05), while values with different small letter superscripts indicate significant differences (*p* < 0.05). A, *p*-value of one-way ANOVA; L, *p*-value of linear analysis; Q, *p*-value of quadratic analysis. The number of samples is 240.

**Table 5 animals-13-01227-t005:** Effects of dietary CF level on the immune function of broilers on day 42.

Item	Crude Fiber Level				SEM	*p*-Value		
	2%	5%	8%	11%		A	L	Q
IgA, μg/mL	157.25 ^d^	179.10 ^c^	219.96 ^a^	199.38 ^b^	5.58	0.013	0.079	0.001
IgG, μg/mL	1214.66 ^b^	1380.83 ^ab^	1691.00 ^a^	1539.16 ^ab^	61.64	0.027	0.163	0.021
IgM, μg/mL	551.01 ^c^	606.26 ^bc^	714.33 ^a^	663.96 ^ab^	17.37	0.010	0.061	0.002
Spleen index, %	0.10	0.12	0.09	0.10	0.01	0.284	0.612	0.535
Fabricius index, %	0.12	0.15	0.14	0.14	0.01	0.596	0.718	0.139

IgA, immunoglobulin A; IgG, immunoglobulin G; IgM, immunoglobulin M. In the same row, values with no letter or the same letter superscripts mean no significant difference (*p* > 0.05), while values with different small letter superscripts indicate significant differences (*p* < 0.05). The number of samples is 64.

**Table 6 animals-13-01227-t006:** Effects of dietary CF level on plasma biochemical indexes of broilers on day 42.

Item	Crude Fiber Level				SEM	*p*-Value		
	2%	5%	8%	11%		A	L	Q
TG, mmol/L	0.75	1.63	2.33	1.06	0.28	0.101	0.222	0.071
T-CHO, mmol/L	9.61 ^a^	7.76 ^ab^	6.43 ^b^	6.00 ^b^	0.44	0.014	0.026	0.054
LDL, mmol/L	1.10	0.92	1.34	0.43	0.09	0.094	0.693	0.433
HDL, mmol/L	2.18	2.15	1.99	2.23	0.06	0.597	0.941	0.371
GLU, mmol/L	5.74 ^a^	5.22 ^b^	4.15 ^d^	4.71 ^c^	0.13	0.020	0.075	0.001
GH, μg/mL	5.73 ^c^	6.42 ^bc^	7.73 ^a^	7.07 ^ab^	0.23	0.010	0.069	0.011
INS, mU/L	24.99	22.96	23.21	21.02	0.82	0.657	0.238	0.507

TG, triglyceride; T-CHO, total cholesterol; LDL, low-density lipoprotein; HDL, high-density lipoprotein; GLU, total blood glucose; GH, growth hormone; INS, insulin. In the same row, values with no letter or the same letter superscripts mean no significant difference (*p* > 0.05), while values with different small letter superscripts indicate significant differences (*p* < 0.05). The number of samples is 64.

**Table 7 animals-13-01227-t007:** Effect of dietary CF level on intestinal morphology of broilers on day 42.

Item		Crude Fiber Level				SEM	*p*-Value		
		2%	5%	8%	11%		A	L	Q
Duodenum	VH, μm	1360.84 ^b^	1374.56 ^b^	1463.26 ^a^	1218.93 ^c^	16.61	0.021	0.233	0.012
	CD, μm	307.87 ^a^	242.35 ^b^	230.23 ^b^	252.73 ^b^	4.90	0.015	0.132	0.017
	V/C	4.87 ^c^	5.46 ^ab^	6.00 ^a^	5.36 ^bc^	0.10	0.023	0.109	0.024
Jejunum	VH, μm	1280.92 ^b^	1472.41 ^a^	1517.28 ^a^	1391.41 ^b^	17.10	0.013	0.766	0.010
	CD, μm	234.09 ^a^	189.88 ^bc^	175.36 ^c^	199.15 ^b^	4.09	0.019	0.140	0.029
	V/C	6.10 ^b^	6.62 ^b^	8.21 ^a^	6.91 ^b^	0.17	0.022	0.110	0.036
Ileum	VH, μm	839.62 ^b^	915.01 ^a^	939.65 ^a^	921.54 ^a^	12.50	0.025	0.219	0.019
	CD, μm	148.36 ^a^	145.52 ^a^	123.21 ^b^	146.74 ^a^	2.30	0.019	0.599	0.011
	V/C	6.29	6.38	6.94	6.34	0.12	0.222	0.991	0.288

VH, villus height; CD, crypt depth; V/C, ratio of VH and CD. In the same row, values with no letter or the same letter superscripts mean no significant difference (*p* > 0.05), while values with different small letter superscripts indicate significant differences (*p* < 0.05). The number of samples is 64.

**Table 8 animals-13-01227-t008:** Effects of dietary CF level on gastrointestinal development of broilers on day 42.

Item		Crude Fiber Level				SEM	*p*-Value		
		2%	5%	8%	11%		A	L	Q
Organ Index, %	Gizzard	1.41	1.42	1.55	1.51	0.03	0.184	0.082	0.125
	Proventriculus	0.40	0.36	0.39	0.41	0.01	0.338	0.978	0.551
Relative Length, cm/kg	Duodenum	15.50	14.11	14.49	15.54	0.26	0.140	0.998	0.632
	Jejunum	33.08	32.03	31.73	34.51	0.51	0.201	0.579	0.176
	Ileum	30.24 ^b^	32.40 ^b^	33.49 ^a^	29.17 ^b^	0.58	0.027	0.768	0.076
	Cecum	15.91 ^c^	17.09 ^b^	19.94 ^a^	17.73 ^b^	0.35	0.011	0.138	0.001
Relative weight, g/kg	Duodenum	5.73	5.54	5.44	5.37	0.10	0.648	0.125	0.312
	Jejunum	11.81	11.65	12.76	11.75	0.23	0.301	0.426	0.708
	Ileum	10.35	8.97	9.71	9.98	0.21	0.143	0.925	0.096
	Cecum	14.71	14.59	14.08	14.87	0.44	0.939	0.523	0.817

In the same row, values with no letter or the same letter superscripts mean no significant difference (*p* > 0.05), while values with different small letter superscripts indicate significant differences (*p* < 0.05). The number of samples is 480.

## Data Availability

The data supporting the reported results and conclusions can be found in the submitted figures and tables. Additional research materials and protocols that are relevant to the study are available from the corresponding author upon reasonable request.

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
