# Peer review of "Dietary Fiber Level Improve Growth Performance, Nutrient Digestibility, Immune and Intestinal Morphology of Broilers from Day 22 to 42"

_animals, 2023, doi:10.3390/ani13071227_

Round 1

Reviewer 1 Report

Paper titled “Effects of Dietary Fiber Level on Growth Performance, Nutrient Digestibility, Immune Function and Intestinal Morphology of Broilers in the Late Rearing Period” by Zhang et al. investigated a very interesting concept in broiler nutrition. Several concern must be however, elucidated.

Specific remarks

1. Lines 19-20. The connection fiber vs. antibiotic-free diet/feeding is not clear here

2. Lines 29-31. This is not clear why dietary treatments were applied from d 21? It is known that proper gut development is essential in the earlier growth-stage in chickens. The same for conclusion – it must be stressed that investigated period was from 21 d of broiler life (which means for broiler chickens from the half-period of rearing).

3.  please specify the sex of the birds

4. Line 149. The immune function in birds is characterized by a complex interaction between number of factors and host response – simple immune organs weighting and selected immunoglobulin concentration determination does not give wide picture

5. Regarding histological analysis, was the other important indices i.e. absorptive area of villi, goblet cells, mucosa thickness determined?

6. Performance response – in fact F/G ratio did not differ between treatments thus, how to justify increased fiber level in the diet, which is associated with the higher cost of feed usually?

7. What, in fact indicate, increased/decreased caecum length? I can accept some correlation between caecum contents (in terms of volume or i.e., bacterial composition) and dietary fiber but its lengths seem to be meaningless

8. Table 4. How to explain that digestibility of CP was higher by 37% and EE by 46% in treatment Fiber 11% vs. fiber 2%, and it did not affect significantly F/G ratio between this groups?     

Author Response

Response to Reviewer 1 Comments

On behalf of all the contributing authors, I would like to express our sincere appreciations of your letter and reviewers’ constructive comments concerning our article entitled “Effects of Dietary Fiber Level on Growth Performance, Nutrient Digestibility, Immune Function and Intestinal Morphology of Broilers in the Late Rearing Period” (animals-2259183). These comments are all valuable and helpful for improving our article. According to the associate editor and reviewers’ comments, we have made extensive modifications to our manuscript to make our results convincing.

Response to reviewer 1

Paper titled “Effects of Dietary Fiber Level on Growth Performance, Nutrient Digestibility, Immune Function and Intestinal Morphology of Broilers in the Late Rearing Period” by Zhang et al. investigated a very interesting concept in broiler nutrition. Several concern must be however, elucidated.

Point 1. Lines 19-20. The connection fiber vs. antibiotic-free diet/feeding is not clear here

Response 1:We are sorry for the inaccurate description. We have changed it. ( see Lines 19-20)

  1. Lines 29-31. This is not clear why dietary treatments were applied from d 21? It is known that proper gut development is essential in the earlier growth-stage in chickens. The same for conclusion – it must be stressed that investigated period was from 21 d of broiler life (which means for broiler chickens from the half-period of rearing).

Response 2: Thanks for your suggestion. We conducted two experiments, one was 22-42d, and the other was 1-42d. The purpose is to explore the fiber requirements of broilers at different growth stages and the impact of fiber at different stages on intestinal development. This manuscript is mainly about 22-42d experimental research. Test data of 1-42d has not been published yet.

  1. please specify the sex of the birds

Response: 3Thanks for your kind suggestion. We have added the sex of the birds to the manuscript. (see Lines 29,112)

  1. Line 149. The immune function in birds is characterized by a complex interaction between number of factors and host response – simple immune organs weighting and selected immunoglobulin concentration determination does not give wide picture

Response 4: Thanks for your kind suggestion. We have replaced immune function with immune index. (see Line 161)

  1. Regarding histological analysis, was the other important indices i.e. absorptive area of villi, goblet cells, mucosa thickness determined?

Response 5: Thank you for your valuable comments. In this study, only part of the intestinal morphology was measured, and the indexes such as the absorption area of villi, goblet cells and mucosal thickness were not measured. We will measure and analysis these indicators in the further research.

  1. Performance response – in fact F/G ratio did not differ between treatments thus, how to justify increased fiber level in the diet, which is associated with the higher cost of feed usually?

Response 6:Thanks for your suggestion. At 29-35 days, the F/G of 5% and 8% CF groups was significantly lower than that of 2% and 11% groups, indicating that 5% - 8% CF could reduce F/G. In addition, there was no significant difference in F/G between different treatment groups at 36-42d, but the F/G at 36-42d was lower than that at 29-35d, indicating that fiber also played an important role at 36-42d. Therefore, considering comprehensively the late feeding period (22-42d), combined with quadratic regression analysis, it was more accurate to suggest that the optimum CF level was 7% for F/G.

  1. What, in fact indicate, increased/decreased caecum length? I can accept some correlation between caecum contents (in terms of volume or i.e., bacterial composition) and dietary fiber but its lengths seem to be meaningless

Response 7:Thank you for kind suggestion. There are also studies that take intestinal length as an indicator of intestinal development (Intestinal Development and Function of Broiler Chickens on Diets Supplemented with Clinoptilolite. Asian-Australasian Journal of Animal Sciences, 2013. Supplementing the early diet of broilers with soy protein concentrate can improve intestinal development and enhance short-chain fatty acid-producing microbes and short-chain fatty acids, especially butyric acid. Journal of Animal Science and Biotechnology,2023.) Intestinal morphology is the main indicator and intestinal length could be the auxiliary reference indicator.

  1. Table 4. How to explain that digestibility of CP was higher by 37% and EE by 46% in treatment Fiber 11% vs. fiber 2%, and it did not affect significantly F/G ratio between this groups?

Response 8:Although the digestibility of nutrients is different, the content of different nutrients was different, and the digestible nutrient content was also different; In addition, nutrients are also used for various metabolism of the body. Therefore, the F/G may be not difference.

Reviewer 2 Report

Manuscript

 Effects of Dietary Fiber Level on Growth Performance, Nutrient Digestibility, Immune Function and Intestinal Morphology of Broilers in the Late Rearing Period

There is some topic that should be improved in this manuscript as below:

Please include the analyzed % of crude fiber in each experimental diet (Table 1) and results tables

Late rearing period in this trial means 22 to 42 d. When we see late we expect these effects in heavier or slow growing broilers. I would recommend use from d 22 to 42 instead of late period

Simple Summary: delete first sentence

 Dietary fiber when in high levels for broiler chickens

L20 – Under

Abstract:

It is too long, it is necessity to simplify and use the most important information

Delete first sentence

When you reduce the length of abstract, please delete background, methods, results, conclusions

L26 delete  Methods: late rearing period.

L27 delete AA

The writing should be improved to increase the flow and reading

L27 and L29 – there is a repeated information about 4 trt

L32 delete  significantly, because you are showing the P-value that explains the same

L33 delete  significantly

L36 delete the (in front of broilers_

Please review when is necessary to use the and when it is not…

Same comment as before for late rearing period

Keywords in alphabetical order

L93 – replace purpose by objective

L96 repeated statement, please delete in the late feeding period to provide a reference for the application of dietary fiber  in broiler production.

MM

Again, please improve increase the flow and reading

L101 - Written informed consent was obtained from the owners for the participation of their animals in this study. This is not clear….

L105 delete again (AA)… this is more usual for amino acids. The genetics doesn’t need abbreviation. Please review if you mentioned this more times

L105 – One-d-old chicks please correct

L114 – replace feces by excreta

Please indicate CF level in the common basal diet fed from d 1 to 21

L155 – delete last sentence The experiment was conducted at 115 the animal husbandry teaching experiment base of Hebei Agricultural University.

L117 - air dry 117 basis, please correct to on dry matter basis

L-Lys not italicized

AP abbreviation is not usual, please specify. Use full words instead of abbreviation

Same for other nutrients and for CaHPO4

L128 replace in the morning, noon and evening by 3 times in one day

L133 3000

L134 – repeated statement At the end of the experiment (42 days of age)

L137 -139  please improve

Acid insoluble ash as an ingredient/marker? It was not celite? And then the Acid insoluble ash was determined

How much was included?

For how long?

Replace indicator by marker

Replace all feces by excreta, pleas review all the text

For all the document and here again 2.5. Immune function

Please review the use of the…

L187-202 – better if you separate information on table 2 and then use the sentences related to table 3 right before table 3

Table 2 The analyzed Crude fiber level can be include close to the formulated levels

Table 3 replace Extremum by optimal

Regression equation need spaces

Please explain what is CP, %, EE, %.... ADF, %  in table 3… this is not clear

Table 4 to 8 – Item instead of Items

 L222  L237 257, 268, and 276…. explain A, L and Q again in the respective footnote

L225 extra “.” . Immune

L277 -302 – very long paragraph

368-394 – same comment… please review

General comments:

Abstract should be improved

Material and methods sometimes is not clear or has repeated statements  

Discussion – there is a bunch of citations, one after another, but you are missing the connection. You are missing the discussion between your data and the results found in the literature, please review

Conclusions please conclude also based in analyzed CF

Author Response

Response to Reviewer 2 Comments

On behalf of all the contributing authors, I would like to express our sincere appreciations of your letter and reviewers’ constructive comments concerning our article entitled “Effects of Dietary Fiber Level on Growth Performance, Nutrient Digestibility, Immune Function and Intestinal Morphology of Broilers in the Late Rearing Period” (animals-2259183). These comments are all valuable and helpful for improving our article. According to the associate editor and reviewers’ comments, we have made extensive modifications to our manuscript to make our results convincing.

Response to reviewer 2

Thanks for your suggestion. We have read the comments of the reviewer 2 carefully and revised and cited manuscript according to comments on a point-by-point basis. There are two questions to be explained especially as the following:

  1. L137 -139 please improve

Acid insoluble ash as an ingredient/marker? It was not celite? And then the Acid insoluble ash was determined

How much was included?

For how long?

Response 1:Acid-insoluble ash (AIA) in feed is insoluble in boiling HCI and is not absorbed by animals. AIA is a reliable endogenous indicator. AIA method is widely used in experiments on nutrient digestibility (Effects of dietary supplementation with itaconic acid on the growth performance, nutrient digestibility, slaughter variables, blood biochemical parameters, and intestinal morphology of broiler chickens. Poultry Sci .2022.Starving for Nutrients: Anorexia During Infection with Parasites in Broilers Is Affected by Diet Composition. Poultry Sci. 2022). The calculation method is as follows:

ATTD (%) = (100−A1/A2×F2/F1×100)

In which A1 represents the AIA content of the feed, A2 represents the AIA content of the feces, F1 represents the nutrient content of the feed, and F2 represents the nutrient content of the feces.

2- L187-202 – better if you separate information on table 2 and then use the sentences related to table 3 right before table 3

Table 2 The analyzed Crude fiber level can be include close to the formulated levels

Response 2:Thanks for your constructive comments. We have separate information on table 2 and then use the sentences related to table 3 right before table 3. (see Line 213-230)

The analytical crude fiber level of this manuscript is very close to the formulated levels.

Manuscript

 Effects of Dietary Fiber Level on Growth Performance, Nutrient Digestibility, Immune Function and Intestinal Morphology of Broilers in the Late Rearing Period

There is some topic that should be improved in this manuscript as below:

Please include the analyzed % of crude fiber in each experimental diet (Table 1) and results tables

Response:(see Line 133)

Late rearing period in this trial means 22 to 42 d. When we see late we expect these effects in heavier or slow growing broilers. I would recommend use from d 22 to 42 instead of late period

Response:(see Line 4)

Simple Summary: delete first sentence

 Dietary fiber when in high levels for broiler chickens

L20 – Under

 Response:(see Line 13,21)

Abstract:

It is too long, it is necessity to simplify and use the most important information

Delete first sentence

When you reduce the length of abstract, please delete background, methods, results, conclusions

L26 delete  Methods: late rearing period.

L27 delete AA

The writing should be improved to increase the flow and reading

L27 and L29 – there is a repeated information about 4 trt

L32 delete  significantly, because you are showing the P-value that explains the same

L33 delete  significantly

L36 delete the (in front of broilers_

Please review when is necessary to use the and when it is not…

Same comment as before for late rearing period

Keywords in alphabetical order

Response:(see Line 25-47)

L93 – replace purpose by objective

L96 repeated statement, please delete in the late feeding period to provide a reference for the application of dietary fiber  in broiler production.

Response:(see Line 98)

MM

Again, please improve increase the flow and reading

L101 - Written informed consent was obtained from the owners for the participation of their animals in this study. This is not clear….

L105 delete again (AA)… this is more usual for amino acids. The genetics doesn’t need abbreviation. Please review if you mentioned this more times

L105 – One-d-old chicks please correct

L114 – replace feces by excreta

Please indicate CF level in the common basal diet fed from d 1 to 21

L155 – delete last sentence The experiment was conducted at 115 the animal husbandry teaching experiment base of Hebei Agricultural University.

L117 - air dry 117 basis, please correct to on dry matter basis

L-Lys not italicized

AP abbreviation is not usual, please specify. Use full words instead of abbreviation

Same for other nutrients and for CaHPO4

Response:(see Line 101-131)

L128 replace in the morning, noon and evening by 3 times in one day

L133 3000

L134 – repeated statement At the end of the experiment (42 days of age)

L137 -139  please improve

Acid insoluble ash as an ingredient/marker? It was not celite? And then the Acid insoluble ash was determined

How much was included?

For how long?

Replace indicator by marker

Replace all feces by excreta, pleas review all the text

Response:(see Line 140-156)

For all the document and here again 2.5. Immune function

Please review the use of the…

L187-202 – better if you separate information on table 2 and then use the sentences related to table 3 right before table 3

Table 2 The analyzed Crude fiber level can be include close to the formulated levels

Table 3 replace Extremum by optimal

Regression equation need spaces

Please explain what is CP, %, EE, %.... ADF, %  in table 3… this is not clear

 Response:(see Line 161,213-235)

Table 4 to 8 – Item instead of Items

 L222  L237 257, 268, and 276…. explain A, L and Q again in the respective footnote

L225 extra “.” . Immune

L277 -302 – very long paragraph

368-394 – same comment… please review

Response:(see Line 245-248,262,280,295,303,250)

Reviewer 3 Report

1-Starting the title with the phrase "the effects of" has become overused. The title should be changed to better reflect the results than just the objectives.

2- The crude fiber content is supposed to reflect the plant cell wall components (including cellulose, hemicellulose, and lignin). However, crude fiber isn't a reliable indicator of the amount of fiber in feeds because it underestimates the real content of the fiber.

3-The Introduction section (especially lines 59-87) reads like a literature review. Previous work on the topic should be briefly summarized without excessive citations that may make it less appealing to the reader. Also, the authors stated at the end that there is not enough data to support this hypothesis. What hypothesis?

4-The sex of the birds used should be specified. More information pertaining to the housing system and rearing conditions is needed. Were the birds housed in floor pens or cages? What was the lighting program? The NRC 1994 requirements are outdated and underestimate the modern fast-growing broilers. Your birds couldn't reach their maximum genetic potential because of feeding low nutrient densities. Breeder companies offer tables for nutrient requirements for their strains. In Table 1, 98.5% after "L-Lys HCl" means lysine or L-Lys HCl percentage. L-Lys HCl is 78.5% Lysine. The reported ME values are kcal/kg, not Mcal/kg. Also, please include NDF and ADF in Table 1.

5-All materials used in this study must have catalog #'s reported in the text. This is essential for reproducibility by other researchers.

6-The feed to gain data in Table 2 doesn't look correct. It is expected that FCR increases with age, but your data show otherwise. Check your data. In general, the FCR data you presented are significantly higher than expected, and the birds couldn't reach their genetic potential in growth based on the Arbor Acres broiler Manual.

7- For the tables, The authors should specify in the title the age in days at which the measurements were taken. Also, footnote the number of samples.

Author Response

Response to Reviewer 3 Comments

On behalf of all the contributing authors, I would like to express our sincere appreciations of your letter and reviewers’ constructive comments concerning our article entitled “Effects of Dietary Fiber Level on Growth Performance, Nutrient Digestibility, Immune Function and Intestinal Morphology of Broilers in the Late Rearing Period” (animals-2259183). These comments are all valuable and helpful for improving our article. According to the associate editor and reviewers’ comments, we have made extensive modifications to our manuscript to make our results convincing.

Response to reviewer 3

1.Starting the title with the phrase "the effects of" has become overused. The title should be changed to better reflect the results than just the objectives.

Response 1: Thanks for your constructive suggestion. We have revised in the manuscript. (see Lines 1-4)

  1. The crude fiber content is supposed to reflect the plant cell wall components (including cellulose, hemicellulose, and lignin). However, crude fiber isn't a reliable indicator of the amount of fiber in feeds because it underestimates the real content of the fiber.

Response 2: Thanks for your suggestion. The standard for the fiber requirements of broilers was lack and it is also our research purpose to study the fiber requirements of broilers. However, the research plan needs scientific reference basis. At present, there are a few reports on the research of CF requirement, and also refering to the approximate range of CF in the commercial feed, so the CF index and corresponding addition level were selected in our study. Thank you for your constructive comments. Now we are also making experiment design that the impact of fiber components (such as soluble fiber and insoluble fiber) on broilers and their requirements in the next study.

3.The Introduction section (especially lines 59-87) reads like a literature review. Previous work on the topic should be briefly summarized without excessive citations that may make it less appealing to the reader. Also, the authors stated at the end that there is not enough data to support this hypothesis. What hypothesis?

Response 3: Thanks for constructive comments. We have made changes about excessive citations in the new manuscript according to suggestions (see Lines 70-71,79-82). In addition, we are sorry that “hypothesis” was used inaccurately and we also revised (see Lines 94-95).

4.The sex of the birds used should be specified. More information pertaining to the housing system and rearing conditions is needed. Were the birds housed in floor pens or cages? What was the lighting program? The NRC 1994 requirements are outdated and underestimate the modern fast-growing broilers. Your birds couldn't reach their maximum genetic potential because of feeding low nutrient densities. Breeder companies offer tables for nutrient requirements for their strains. In Table 1, 98.5% after "L-Lys HCl" means lysine or L-Lys HCl percentage. L-Lys HCl is 78.5% Lysine. The reported ME values are kcal/kg, not Mcal/kg. Also, please include NDF and ADF in Table 1.

Response 4: Thanks for your kind suggestion. We have revised in the new manuscripts (see Lines 121-126). The latest version of NRC for broilers is 1994, and the nutritional requirements are not much different from the current feeding standards. Many studies also refer to NRC (1994),such as the two studies(Protective effects of Fagopyrum dibotrys on oxidized oil-induced oxidative stress, intestinal barrier impairment, and altered cecal microbiota in broiler chickens. Poultry Science. 2023. Influence of dietary crude protein on growth performance and apparent and standardized ileal digestibility of amino acids in corn-soybean meal-based diets fed to broilers. Poultry Science.2023.). In addition, 98.5% after "L-Lys HCl" means L-Lys HCl percentage.

5.All materials used in this study must have catalog #'s reported in the text. This is essential for reproducibility by other researchers.

Response 5: Thanks for your constructive comments. We have revised in the new manuscript. (see Lines 171-174)

6.The feed to gain data in Table 2 doesn't look correct. It is expected that FCR increases with age, but your data show otherwise. Check your data. In general, the FCR data you presented are significantly higher than expected, and the birds couldn't reach their genetic potential in growth based on the Arbor Acres broiler Manual.

Response 6: Thanks for your suggestion. The reason may be that the proper amount of fiber can improve the intestinal development, digestion and absorption of broilers, so F/G did not increase with the age of the day in this study, but decreased in the third week, which is possible that the fiber effect will be obvious for F/G in the third week. We also carried out another experiment on the effect of fiber on broilers in 1-42 d. The same phenomenon also occurred. It was found that the F/G was also descend after two weeks of feeding (This part of data has not been published yet).

  1. For the tables, The authors should specify in the title the age in days at which the measurements were taken. Also, footnote the number of samples.

Response 7: Thanks for your kind suggestion. We have revised in the new manuscript. (see Lines 213,217-218,244,248,261,263-264,279,282,294,296,302,304)

Round 2

Reviewer 1 Report

Although most of my concerns has been elucidated, without strong and credibility explanation (also in the manuscript text) of remark no. 8, it can not be accepted. It is unlikely in broiler chicekns that recorded changes (digestibility of CP was higher by 37% and EE by 46% in treatment) did not ffect growth of the birds. Strongly suggest verify calculations.

Author Response

Response to reviewer 1

Although most of my concerns has been elucidated, without strong and credibility explanation (also in the manuscript text) of remark no. 8, it can not be accepted. It is unlikely in broiler chickens that recorded changes (digestibility of CP was higher by 37% and EE by 46% in treatment) did not affect growth of the birds. Strongly suggest verify calculations.

Response:

Thanks for your kind suggestion. We have verified the data and calculation, and there is no problem. The data reported in some literature also indicated that the changes in nutrient digestibility did not affect growth performance in broilers (Hanseo Ko et al. Effects of metabolizable energy and emulsifier supplementation on growth performance, nutrient digestibility, body composition, and carcass yield in broilers, Poultry Science,2023. A. R. Zhang et al. Effects of feeding solid-state fermented wheat bran on growth performance and nutrient digestibility in broiler chickens, Poultry Science,2022. Effect of different doses of phytase and protein content of soybean meal on growth performance, nutrient digestibility, and bone characteristics of broilers, Poultry Science, 2021. Xin Zhu et al. Effects of dietary supplementation with itaconic acid on the growth performance, nutrient digestibility, slaughter variables, blood biochemical parameters, and intestinal morphology of broiler chickens, Poultry Science,2022.). The explanation for the results is presumed that nutrients are involved in various metabolic pathways in animals, so high nutrient digestibility may not necessarily improve growth performance. And we will explore and verify the reason in the next step.

Reviewer 2 Report

In general, the responses were very confused and they were not indicated separately.

Acid insoluble ash as an ingredient/marker? It was not celite? And then the Acid insoluble ash was determined

How much was included?

For how long?

Besides the author tried to explain what is AIA marker, he did not include the amount and period.

L213 - Table 2. Effects of dietary CF level on growth performance of broilers on day 22-42.

Should be from day 22 to 42 not on day 22-42.

L129- Table 1. Ingredient composition and nutrient levels of experimental diets during 22-42 days

Better if you also use from day 22 to 43

Table 2 – number of units for F/G should be 3

Table 4 – digestibility. Numbers of units should be 1. It is more difficult to see the data with 2 units. Please explain why the digestibility is too low. Even the CP and EE digestibility in the optimal level is low. Explain and present some data from the literature

Please improve conclusions. It is not clear

Author Response

Response to reviewer 2

1.Acid insoluble ash as an ingredient/marker? It was not celite? And then the Acid insoluble ash was determined

How much was included?

For how long?

Besides the author tried to explain what is AIA marker, he did not include the amount and period

Response:

Thanks for your constructive suggestion. We are sorry that we did not accurately understand your question before. We added the celite to the diets, but we did not describe it clearly in the manuscript before. And we have revised the description in the manuscript. (see Lines 139-143)

In order to determine in vivo nutrient digestibility, On day 40, 2 birds with body weight close to the mean of each group were randomly selected for digestive trial. The acid-insoluble ash (AIA) marker technique was used to measure nutrient digestibility throughout day 40-42. On day 40 (3 d before excreta collection), an additional source of AIA with celite was supplied to the diet at a dosage of 10 g/kg.

2.L213 - Table 2. Effects of dietary CF level on growth performance of broilers on day 22-42.

Should be from day 22 to 42 not on day 22-42.

L129- Table 1. Ingredient composition and nutrient levels of experimental diets during 22-42 days

Better if you also use from day 22 to 43

Response:

Thank you for kind suggestion. We have changed it. ( see Lines 218,129)

3.Table 2 – number of units for F/G should be 3

Table 4 – digestibility. Numbers of units should be 1. It is more difficult to see the data with 2 units.

Response:

Thank you for kind suggestion. We have changed it. (see Table 2 and Table 4)

  1. Please explain why the digestibility is too low. Even the CP and EE digestibility in the optimal level is low. Explain and present some data from the literature

Response:

Thank you for kind suggestion. On the one hand, the commercial diet fed in this experiment from 1 to 21d, and the experimental diets were fed from 22 to 42d. The experimental diets with changing in ingredients were a stress to broilers, which may affect nutrient digestibility for only three weeks. If feeding time is prolonged, different results may be obtained. Because we also conducted another experiment on the effect of fiber levels on broilers from 1 to 42 days, and we found that the nutrients digestibility were higher than the data obtained from the experiment with 22 to 42 day. This needs further exploration. On the other hand, there are polar and non-polar groups in fiber molecules, forming polar and non-polar regions in the spatial structure, so that it can be combined with water molecules, but also with the lipid interface. When entering the digestive tract of animals, it is easy to wrap fat particles and glycogen molecules, and then affecting the digestion and absorption of nutrients (Ding Ranran. Effects of Insoluble Fiber on Growth Performance, Intestinal Health and Liver Lipid Metabolism of Broilers[D]. Master’s Thesis. Zhengzhou: Henan Agricultural University,2022.In Chinese).

5.Please improve conclusions. It is not clear

Response:

Thank you for kind suggestion. We have changed it. (see Lines 449-453)

Reviewer 3 Report

my comments have been taken care of 

Author Response

Thank you very much for your comments and suggestions.